# High prevalence of cardiovascular risk factors in pregnant women in Benin

**Eyram Maria Concheta Tchibozo**[1,2,3,4☯], **Yessito Corine Houehanou Sonou**[4,5☯],
**Salmane Ariyoh Amidou**[4☯], **Fabrice Hountondji**[4], **Femi Zantou**[4],
**Philippe Lacroix**[1,2,3,6☯]*, **Dismand Stephan Houinato**[1,2,3,4☯], **Holy Bezanahary**[1,2,3,7☯]

1 University Limoges, EpiMaCT—Epidemiology of Chronic Diseases in Tropical Zone, Institute of Epidemiology and Tropical Neurology, OmegaHealth, Limoges, France, 2 Inserm, U1094, EpiMaCT—Epidemiology of Chronic Diseases in Tropical Zone, Limoges, France, 3 IRD, U270, EpiMaCT—Epidemiology of Chronic Diseases in Tropical Zone, Limoges, France, 4 Faculty of Health Sciences, Epidemiology Laboratory of Chronic and Neurologic Diseases, University of Abomey-Calavi, Cotonou, Benin, 5 National School of Public Health (ENATSE), University of Parakou, Parakou, Benin, 6 Department of Vascular Surgery and Vascular Medicine, Dupuytren University Hospital, Limoges, France, 7 Department of Internal Medicine, Dupuytren University Hospital, Limoges, France

☯ These authors contributed equally to this work.
* Philippe.lacroix@unilim.fr

**Data Availability Statement:** All relevant data are within the manuscript and its Supporting Information files.

## Abstract

### Introduction

Modifiable cardiovascular risk factors (CVRF) are highly prevalent in SubSaharan African communities. In these countries the burden of CVRF during early pregnancy has been poorly documented.

### Aim

The objective of this study was to describe the frequency of CVRF in pregnant women before the 20th week of gestation in Benin

### Methods

Consecutive pregnant women with a gestational age < 20th week were included in 30 maternity clinics in Benin. Univariate and multivariate analyses were used to determine characteristics associated with CVRF.

### Results

1244 pregnant women were included (680 (54.7%) in urban areas and 584 (45.3%) in rural areas). The median age was 26 years. The frequencies of high blood pressure (HBP), obesity and diabetes were 18.9%, 15.0% and 3.1% respectively. Very few women (25.3%) were aware of the HBP disorder. HBP was associated with an age $\geq$ 35 years (OR = 1.7, 95% CI:1.1–2.7), a rural setting (OR = 2.6; 95%CI:1.9–3.5), an insufficient consumption of fruits and vegetables (OR = 3.2; 95%CI:2.0–5.3) and a history of at least 2 fetal losses (OR = 1.9; 95% CI [1.4–2.7]). The risk of being overweight was associated with an age >24 years old (OR = 1.6; 95%CI:1.1–2.2) conversely a rural setting was protective (OR = 0.7; 95%CI:0.5–

**Funding:** The author(s) received no specific funding for this work.

**Competing interests:** The authors have declared that no competing interests exist.

0.9). Obesity was associated with an age > 35 years old (OR = 4.1; 95%CI:2.5–6.8) and a rural setting (OR = 0.3; 95%CI: 0.2–0.5).

## Conclusion

The frequency of CVRF in women before 20th week of gestation was high. Most of the women were unaware of the disorder. Thus the screening of CVRF among women of reproductive age might be relevant.

## Introduction

Cardiovascular risk factors (CVRF) are highly prevalent in Sub Saharan African communities. In the TAHES study which was conducted in a rural setting in Benin, prevalence of high blood pressure and obesity in the female population were up to 32% and 12% respectively [1]. In Sub Saharan Africa (SSA) diabetes prevalence ranged from 1.6% to 21.6%; these results are related to different methodologies used to study diabetes prevalence; however in many studies a higher prevalence in women than in men was described [2]. In a large meta-analysis chronic hypertension, pregestational diabetes and a body mass index (BMI) > 30 were associated with an increased risk of pre-eclampsia (PE) [3]. In another review limited to SSA studies hypertension, diabetes, obesity and alcohol consumption were significantly associated with a risk of PE [4].

Reducing maternal mortality remains a global public health challenge [5], particularly in SSA. Sustainable Development Goal 3 aims to reduce the global maternal mortality rate to below 70 per 100,000 live births by 2030. According to the World Health Organization (WHO), approximately 830 women die every day worldwide due to pregnancy or childbirth-related complications. Most of these deaths (99%) occur in developing countries, two thirds in SSA [6]. In 2017, the estimated maternal mortality rate in SSA was 542 per 100,000 live births [7]. In Benin, the maternal mortality rate was 405 per 100,000 live births in 2015, and PE accounted for 22% of these deaths [8]. In 2016, 16.5% of pregnancies were complicated by PE [9].

Hypertension or diabetes detected in the first trimester of pregnancy are related to preexisting chronic disorders [10, 11]. Early screening and management of CVRF during pregnancy is recommended in order to reduce maternal morbidity and mortality [12].

Very few studies focused on the frequency of CVRF in pregnant women in SSA. The aim of this study was to describe CVRF frequencies in women before the 20th gestational week (GW) in Benin.

## Materials and methods

### Study design, population, recruitment, data collection

A cross-sectional prospective study was conducted from 1st February 2020 to 30th November 2021 in 30 selected maternity clinics distributed across Benin in rural and urban districts. The district, then the maternity clinics in each district were randomly selected. All consecutive women attending antenatal care (ANC) clinics on the same day were included. Inclusion criteria were: age ≥ 15 years old and pregnancies < 20 GW. The gestational age was assessed by the midwives. Informed consent was required. Deaf-mute and disabled women who were unable to answer the questionnaires were excluded. The sample size was calculated on the basis of expected prevalence of high blood pressure among women (28.4%) based on previous studies

in the community in Benin [13]. Selecting a 3% precision level, a 5% type I error level, and non-response rate of 10%, a sample size of 1147 subjects was found to be mandatory. Recruitment of 1244 subjects was provided.

Data collection was carried out by trained interviewers and midwives. In addition, patient medical records were reviewed. Sociodemographic and economic data were age, marital status, education level, residency (urban or rural) according to Benin's Statistical and Economic Analysis Institute (INSAE) list [14]. Fruits and vegetables intake, salt and alcohol intakes were according to the WHO STEPS survey guidelines [15]).Clinical data were blood pressure, diabetes and the associated treatments. Obstetrical data consisted of gestity, parity, past medical history of fetal loss, gestational hypertension, PE, gestational diabetes, macrosomia.

Blood pressure was measured using an electronic device (Omron ® M3, Omron Healthcare, Japan) three times on a single arm after 15 minutes of rest, with an interval of 3 minutes between measurements. The blood pressure status was defined based on the average of the last two measurements. High blood pressure (HBP) was defined [10] as systolic blood pressure $\geq$ 140 mm Hg and/or diastolic blood pressure $\geq$ 90 mm Hg or being on antihypertensive treatment.

Capillary blood glucose was measured using glucometers (Accu-Chek® Active, Roche, Switzerland). The women were fasting for at least 8 hours. Otherwise, a fasting capillary blood glucose measurement was measured the next day. According to International Association of Diabetes and Pregnancy Study Groups (IADPSG) recommendations, gestational diabetes was defined as fasting capillary blood glucose $\geq$ 0.92 g/l (5.1 mmol/l) and pre-existing diabetes or overt diabetes was defined as fasting capillary blood glucose $\geq$ 1.26 g/l (7 mmol/l) [11] or being on antidiabetic treatment. The electronic devices were calibrated and their accuracy was verified before data collection.

Brachial circumference (BC) was measured using a measuring tape at the midpoint between the tip of the elbow and the acromion on the non-dominant arm of the pregnant women. Overweight was defined as BC $\geq$ 27 cm and obesity as BC $\geq$ 31 cm [16]. Definitions of alcohol consumption, tobacco use and fruit and vegetable intakes were based on WHO criteria. Fruit and vegetable low intakes was defined as consumption < 5 servings per day.

## Data management and statistical analysis

The data were collected and entered on tablets using the KoboCollect application. They were analyzed using the R software (R Core Team, 2023). The Shapiro-Wilk test was used to assess the normality of the distribution of quantitative variables. Quantitative variables were described using medians and quartiles and qualitative variables with frequencies and proportions. Proportions were compared between two groups using the Chi-square test or the Fisher test. A significant difference was noted with a p-value < 5%.

After univariate analysis, factors associated with CVRF were identified using stepwise descending multivariable logistic regression. The missing data due to non-responses were not included in the models. The threshold for variable retention in models was set at 20%. Model validation was performed using the Hosmer-Lemeshow test. Adjusted odds ratios to age and area with 95% confidence intervals were used as measures of association, with a significance level set at a p-value < 5%. The analysis and presentation of results of statistical analysis are in accordance with the STROBE recommendations.

## Ethics approval and consent to participate

The study was approved by the ethics and health sciences research committee of Parakou University (Benin) on 08th January 2020 (REF:0594/CLERB-UP/P/SP/P/SA). All experiments

were performed in accordance with relevant guidelines and regulations. All the subjects gave a verbal consent before inclusion. The consent was verbal due to a large number of illiterate women and it was documented in the database. If the pregnant women was minor, the consent was obtained from relatives.

## Results

Overall, 3533 pregnant women were screened and 1244 met the inclusion criteria of pregnancy < 20 GW (680 (54.7%) living in urban areas and 584 (45.3%) in rural areas). The median age was 26 years (interquartile range: 22–30). The age group of 25 to 35 years was the most represented (48.6%). Over three-quarters of the women were in a marital relationship (94.2%). In rural areas, most women had no formal education (36.2%). More than half of the women had a monthly income of less than 40,000 FCFA (61 € or 65 $) (79.9%). The general characteristics are displayed on Table 1.

The CVRF frequencies are displayed on Table 2. Among 1235 women with blood pressure collection, 205 (16.6%) had abnormal pressures including 31 on treatment. 28 (2.3%) women who were on treatment before the study had normal blood pressure. Overall, 233 (18.9%) had HBP. In the abnormal pressure group, 174 (74.7%) were unaware of being hypertensive.

Capillary blood glucose levels were measured in 1013 (81.2%) subjects. Overall, 32 (3.2%) were considered diabetic, 12 were treated and 20 non-treated had glucose levels ≥ 1.26 g/l. Regarding the last group 18 were unaware of the disorder. Considering the threshold of

**Table 1. Global sociodemographic characteristics of the women according to area.**

| | General (N = 1244) | Urban (N = 680) | Rural (N = 564) | |
|---|---|---|---|---|
| | n (%) | n (%) | n (%) | P |
| **Age (years)** | | | | **P < .01** |
| 15–24 | 500 (40.2) | 252 (37.1) | 248 (44.0) | |
| 25–34 | 605 (48.6) | 359 (52.8) | 246 (43.6) | |
| ≥ 35 | 139 (11.2) | 69 (10.1) | 70 (12.4) | |
| **Formal education** | | | | **P < .001** |
| No schooling | 377 (30.3) | 173 (25.4) | 204 (36.2) | |
| Primary | 307 (24.7) | 167 (24.6) | 140 (24.8) | |
| Secondary | 454 (36.5) | 254 (37.4) | 200 (35.5) | |
| University | 106 (8.52) | 86 (12.6) | 20 (3.5) | |
| **Marital relashionship** | 1272 (94.2) | 623 (91.6) | 549 (97.3) | |
| **Occupation** | | | | **P < .001** |
| Housekeeper | 267 (21.5) | 136 (20.0) | 131 (23.2) | |
| Retailer | 433 (34.8) | 217 (31.9) | 216 (38.3) | |
| Farmer | 39 (3.1) | 6 (0.9) | 33 (5.9) | |
| Civil servant | 105 (8.4) | 78 (11.5) | 27 (4.8) | |
| Artisans | 248 (19.9) | 149 (21.9) | 99 (17.6) | |
| Student/ Apprentice | 72 (5.8) | 59 (8.7) | 13 (2.3) | |
| Others | 80 (6.4) | 35 (5.1) | 45 (8.0) | |
| **Monthly income**[a] (n = 819) | | | | **P < .001** |
| < 40 000 | 552 (67.4) | 265 (57.6) | 287 (79.9) | |
| 40 000–80 000 | 209 (25.5) | 154 (33.5) | 55 (15.3) | |
| > = 80 000 | 58 (7.1) | 41 (8.9) | 17 (4.7) | |

[a] Franc of the Financial African Community

**Table 2. Global behaviour and cardiovascular characteristic of the women.**

| | General (n = 1244) | Urban (n = 680) | Rural (n = 564) | |
|---|---|---|---|---|
| | n (%) | n (%) | n (%) | P |
| **HBP[a]** (n = 1235) | 233 (18.9) | 90 (13.3) | 143 (25.7) | < .001 |
| **Hyperglycemia** (n = 1013) | 129 (12.8) | 73 (10.8) | 56 (16.7) | .0118 |
| **Diabetes** (n = 1013) | 32 (3.2%) | 23 (3.4%) | 9 (2.7%) | 0,6454 |
| **Overweight** (n = 1240) | 339 (27.3) | 236 (34.8) | 103 (18.3) | < .001 |
| **Obesity** (n = 1240) | 186 (15.0) | 140 (20.6) | 46 (8.2) | < .001 |
| **Tobacco before pregnancy** | 24 (1.9) | 10 (1.5) | 14 (2.5) | .2782 |
| **Tobacco during pregnancy** | 18 (1.6) | 6 (0.9) | 12 (2.1) | .113 |
| **Alcohol before pregnancy** | 432 (34.7) | 272 (40.0) | 160 (28.4) | < .001 |
| **Alcohol during pregnancy** | 278 (22.4) | 153 (22.5) | 125 (22.2) | .9413 |
| **Fruit and vegetables intake [b]** | | | | < .001 |
| < 5 portions | 1002 (80.5) | 598 (87.9) | 404 (71.6) | |
| ≥ 5 portions | 242 (19.6) | 82 (12.1) | 160 (28.4) | |
| **Salt intake according to women [b]** | | | | < .01 |
| Too much | 137 (11.0) | 90 (13.2) | 47 (8.3) | |
| Right or too little | 1069 (85.9) | 574 (84.4) | 495 (87.8) | |
| Unknown | 38 (3.1) | 16 (2.4) | 22 (3.9) | |

[a]HPB: High Blood Pressure

[b]: declarative data

0.92 g/l, 122 women without treatment were above the limit. Among 1240 women, 339 (27.3%) were overweight and 186 (15%) were obese. The majority of women had an insufficient consumption of fruits and vegetables (80.5%).

The obstetrical characteristics are displayed in Table 3. Primigravidae (31%) and nulliparous women (35.1%) were more represented in urban areas.

In the multivariate analysis, HBP was associated with an age ≥ 35 years (OR = 1.7, 95% CI:1.1–2.7), a rural setting (OR = 2.6, 95%CI:1.9–3.5), an insufficient intake of fruits and vegetables (OR = 3.2, 95%CI:2.0–5.3) and a history of at least 2 fetal losses (OR = 1.9, 95%CI [1.4–2.7]) (Table 4).

The risk of being overweight was associated with an age > 24 years (OR = 1.6; 95%CI:1.1–2.2) conversely a rural setting was protective (OR = 0.7; 95%CI:0.5–0.9). Obesity was also associated with an age > 35 years (OR = 4.1; 95%CI:2.5–6.8) and a rural setting was also protective (OR = 0.3; 95%CI: 0.2–0.5) (see S1 Table). Diabetes and hyperglycemia were not associated with any variable (see S1 Table).

## Discussion

A high prevalence of CVRF was documented in this study. The collection of CVRF was limited to women before the 20[th] GW so these CVRF were considered to be chronic, preexisting to the pregnancy. They have to be screened and treated before the pregnancy in order to prevent cardiovascular events not only during the pregnancy but also on long term [17, 18].

Previous studies conducted in Benin general population [1, 13] showed an alarming prevalence of HBP above 28%. In our study including young women the prevalence was up to 18.9%. Few studies in high income countries reported lower prevalence in pregnant women before the 20[th] GW. In a US cohort involving over 18 000 pregnant women the frequency of

**Table 3. Obstetrical characteristics of the women.**

| | General (N = 1244) | Urban (N = 680) | Rural (N = 564) | |
|---|---|---|---|---|
| | n (%) | n (%) | n (%) | P |
| **Gestity** | | | | .1135 |
| 1 | 362 (29.1) | 211 (31.0) | 151(26.8) | |
| > 1 | 882 (70.9) | 469 (69.0) | 413 (73.2) | |
| **Parity** | | | | .0814 |
| 0 | 410 (33.0) | 239 (35.4) | 171 (30.3) | |
| ≥ 1 | 834 (67.0) | 441 (64.9) | 393 (69.7) | |
| **Foetal loss** | | | | < .01 |
| 0–1 | 988 (79.4) | 560 (82.4) | 428 (75.9) | |
| ≥2 | 256 (20.6) | 120 (17.6) | 136 (24.1) | |
| **History of hypertension** (n = 1174) | | | | .8247 |
| Yes | 60 (5.1) | 29 (4.7) | 31 (5.5) | |
| No | 1087 (92.6) | 570 (93.0) | 517 (92.2) | |
| Unknown | 27 (2.2) | 14 (2.1) | 13 (2.3) | |
| **History of pre-eclampsia** (n = 991) | | | | **P < .01** |
| Yes | 67 (6.8) | 43 (8.0) | 24 (5.3) | |
| No | 814 (82.1) | 420 (78.5) | 394 (86.4) | |
| Unknown | 110 (11.1) | 72 (13.5) | 38 (8.3) | |

hypertension before the 20[th] GW was 6.8% [19]. In another North American study the frequency of hypertension was up to 7.3% among African-American women [20]. Very few women were treated and a large majority were unaware of the disorder. Such a characteristic was previously documented in epidemiological studies conducted in SSA [17, 20, 21].

**Table 4. Factors associated with high blood pressure in pregnant women.**

| | Univariate | | | Multivariate | |
|---|---|---|---|---|---|
| | n(%) | OR [95%CI] | P | OR* [95%CI] | P* |
| **Age (years)** | | | < .001 | | |
| 15–24 | 74(14.9%) | 0.73[0.5–0.1] | .050 | 0.8[0.5–1.2] | .386 |
| 25–34 | 117(19.5%) | 1 | - | | |
| ≥ 35 | 42(30.4%) | 1.8[1.2–2.7] | < .01 | 1.7[1.1–2.7] | .015 |
| **Area** | | | < .001 | | |
| Urban | 90(13.3%) | 1 | - | | |
| Rural | 143(25.7%) | 2.3[1.7–3.0] | < .001 | 2.6[1.9–3.5] | < .001 |
| **Fruit and/or vegetables intake** | | | < .001 | | |
| < 5 portions | 22(9.1%) | 2.7[1.7–4.4] | < .001 | 3.2[2.0–5.3] | < .001 |
| ≥ 5 portions | 211(21.2%) | 1 | - | | |
| **Gestity** | | | < .001 | | |
| 1 | 45(12.5%) | 1 | - | | |
| > 1 | 188(21.5%) | 1.9[1.4–2.8] | < .001 | 1.4[0.9–2.1] | .139 |
| **Fetal loss** | | | < .001 | | |
| 0–1 | 253(15.6%) | 1 | - | | |
| ≥2 | 80(31.5%) | 2.49[1.8–3.4] | < .001 | 1.9[1.4–2.7] | < .001 |

*adjusted to age and area

Prevalence of diabetes was close to those documented in the community [13]. However it is important to stress that 12.8% of the women were above the limit of gestational diabetes. In South Africa, a prevalence of 25.8% [22] was reported in a cohort study including 554 women before 26 weeks of gestation.

We focused on pre-pregnancy obesity (15.0%), which predisposes to hypertensive complications, as demonstrated by several studies. Given the context (cross-sectional study, population with low education levels, living in rural African settings for some subjects), very few women were able to accurately report their weight before pregnancy. That is the reason why we opted for measuring the BC before 20 weeks of gestation. Brachial circumference is considered to be a relatively stable measurement during pregnancy among women in developing countries [23]. In Uganda a BC > 28 cm was documented in 66.7% of women without pre-eclampsia [24]. In Ethiopia, the frequency of obesity (BC> 25 cm.) among hypertensive pregnant women was 49%, compared to 35% among non-hypertensive pregnant women [25]. The different threshold definitions for obesity could also explain the differences. In a Mexican survey, pre-pregnancy obesity based on BMI, obesity rate was up to 16%, [26]. In another study conducted in Ghana, in a cohort of pregnant women included before 20 weeks of gestation the rate of obesity (based on BMI) was 33% [27]. Our results are in accordance with the Beninese general population data, where the obesity rate among women was 10.1% [28].

HBP was more prevalent in rural setting. Hinkosa et al. in Ethiopia in 2020 in a case-control study in a referral hospital [29] found that living in rural areas increase the risk of developing hypertension in pregnant women. These results are in accordance with the previous studies conducted in Beninese communities [13]. HBP was also associated with a history of miscarriage. This might suggest that hypertensive complications occurred during the previous pregnancy. Conversely overweight or obesity were associated with an urban setting. Such results are in accordance with other SSA studies in Ghana [30] and Burkina Faso [31]. This could be explained by the nutritional transition but also by the sedentary lifestyle in an urban environment [32]. A significant difference between socio-demographic and economic characteristics was found. The frequencies of pregnant women with a young age, a low level of education and a low level of income were higher in rural than in urban areas. All these data suggest a need for tailored intervention and prevention strategies depending on the setting.

## Strengths and limitations

Blood pressure status was determined on the basis of a single episode. Even if blood pressure should be repeated to confirm true hypertension, we selected the method usually recommended in epidemiological studies in which three measurements were used for the diagnosis of HBP. Fasting capillary blood glucose status was determined after a single test, but we ensured that fasting was respected in order to limit bias. We selected IADPSG definition of gestational diabetes as a CVRF as numerous studies agree it is a risk factor of fetal complication but also preeclampsia [11] and furthermore it predisposes to future cardiometabolic disorders to both woman and offspring. Length of residence is a confounding factor in our analyses. We were unable to collect this information, however in the Beninese context, pregnant women do not move around a great deal from one place to another. The type of study was appropriate in view of the objectives set. It was carried out in a large sample of maternity units, including the Benin referral hospitals. The women were included consecutively and exhaustively. In Benin, according to the latest national surveys, 48% of women of childbearing age did not make at least 4 prenatal visits during pregnancy, whereas WHO recommendations stipulate a minimum of 8 visits. This could introduce a sampling bias in this study. Even if the results may reflect the situation in the general population, the generalization of the results to the whole country calls for caution.

## Conclusions

The frequencies of pre-existing CVRF during pregnancy in this study were significant and differed from living areas. Women were frequently unaware of the disorder. This calls for early use of antenatal care clinics during pregnancy, which should normally be carried out (before 16[th] GW) to detect these CVRF as early as possible in order to limit the risk of complications for the mother and child.

## Supporting information

**S1 Table. Factors associated with hyperglycaemia and obesity in pregnant women.** (DOCX)

## Acknowledgments

The authors thank all the midwives, physicians, investigators and authorities involved in data collection and the participating women.

## Author Contributions

**Conceptualization:** Philippe Lacroix, Dismand Stephan Houinato, Holy Bezanahary.

**Formal analysis:** Eyram Maria Concheta Tchibozo, Holy Bezanahary.

**Investigation:** Eyram Maria Concheta Tchibozo, Fabrice Hountondji, Femi Zantou.

**Methodology:** Eyram Maria Concheta Tchibozo, Yessito Corine Houehanou Sonou, Salmane Ariyoh Amidou, Philippe Lacroix, Dismand Stephan Houinato, Holy Bezanahary.

**Supervision:** Philippe Lacroix, Dismand Stephan Houinato, Holy Bezanahary.

**Validation:** Holy Bezanahary.

**Visualization:** Yessito Corine Houehanou Sonou, Philippe Lacroix, Dismand Stephan Houinato, Holy Bezanahary.

**Writing – original draft:** Eyram Maria Concheta Tchibozo, Philippe Lacroix, Holy Bezanahary.

**Writing – review & editing:** Eyram Maria Concheta Tchibozo, Yessito Corine Houehanou Sonou, Salmane Ariyoh Amidou, Philippe Lacroix, Dismand Stephan Houinato, Holy Bezanahary.

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
