## [Decision Letter · Decision Letter 0]

27 May 2024

PONE-D-24-09850High prevalence of cardiovascular risk factors in pregnant women in BeninPLOS ONE

Dear Dr. Lacroix,

Thank you for submitting your manuscript to PLOS ONE. After careful consideration, we feel that it has merit but does not fully meet PLOS ONE’s publication criteria as it currently stands. Therefore, we invite you to submit a revised version of the manuscript that addresses the points raised during the review process.

We look forward to receiving your revised manuscript.

Kind regards,

Yoshihiro Fukumoto

Academic Editor

PLOS ONE

Journal Requirements:

Reviewers' comments:

Reviewer's Responses to Questions

**Comments to the Author**

1. Is the manuscript technically sound, and do the data support the conclusions?

Reviewer #1: Yes

Reviewer #2: Partly

2. Has the statistical analysis been performed appropriately and rigorously? 

Reviewer #1: I Don't Know

Reviewer #2: Yes

3. Have the authors made all data underlying the findings in their manuscript fully available?

Reviewer #1: Yes

Reviewer #2: No

4. Is the manuscript presented in an intelligible fashion and written in standard English?

Reviewer #1: Yes

Reviewer #2: Yes

5. Review Comments to the Author

Reviewer #1: Thank you for an interesting manuscript! The study focuses on modifiable cardiovascular risk factors (CVRF) among pregnant women in Benin, a country in Sub-Saharan Africa. These risk factors include conditions like high blood pressure (HBP), obesity, and diabetes. The research aims to understand the prevalence of these risk factors before the 20th week of gestation. The study revealed that cardiovascular risk factors (CVRF) were common among pregnant women. High blood pressure (HBP) was associated with age >35 years, rural residency, and insufficient fruit and vegetable intake. Overweight was also linked to age >24 years and rural setting was protective while obesity was higher in women >35 years and rural setting was protective.

1. Can you please clarify the univariate and multivariate regression analysis and the variables that are used in the mutlivariate regression analysis? Perhaps a stepwise multivariate regression analysis would be a good approach.

2. Can you please add discussion about limitations such as sampling bias into this study? Approx., how many pregnant women in Benin fails to visit the antenatal clinic and are thus not included in this study?

3. Can you please clarify why you had no follow-up time and why you specifically set the gestational week to be less than 20?

4. Please include ethic approval numbers and dates.

5. "Blood pressure was measured by an electronic device"- for all women? None were measured manually, perhaps for instance in the setting of atrial fibrillation? Was the electrinic devices calibrated and checked for accuracy?

5. Conclusion: The frequency of CVRF in women before 20th week of gestation was high. Can you please rewrite and avoid "high" since it is subjective.

Reviewer #2: Reviewer’s comments.

This study investigates the prevalence of cardiovascular risk factors among pregnant women under 20 weeks of gestation in general African communities. It was found that the risk factors varied depending on the residence of the pregnant women. Additionally, it is an important issue that many pregnant women with cardiovascular risks such as hypertension and obesity are unaware of their hypertension. To reduce the mortality rates of mothers and fetuses, it is desirable to identify these risk factors early and provide appropriate interventions for pregnant women. While the purpose of this study is understandable, several points can be considered insufficient. These are outlined below.

Major comments:

1. Page 6; line 119: What is the basis for the definitions of overweight and obesity in Brachial circumference?

2. The author should mention the difference between urban and rural backgrounds in the results of Table 1.

3. Page 9; line 157: Which is more accurate to say that the patient had normal blood pressure but was receiving treatment or that the patient had normal blood pressure because of the treatment?

4. In Table 2, the definition of the classification of salt intake should be clarified as to how many grams of salt or sodium it is equivalent to.

5. In Table 4, it is necessary to indicate which confounding factors were adjusted for.

6. In Table 4, it is necessary to show the relationship between overweight, obesity, smoking, and alcohol consumption as factors associated with hypertension in pregnant women.

7. Author should mention limitations and strengths in the discussion.

Minor comments:

1. Page 9; line 157: Twenty-eight women (2.3%)→28 (2.3%)

2. Page 9; lines 161-163: Add (%) after the number of subjects.

3. In Table 4, the reference should be shown for each variable.

6. PLOS authors have the option to publish the peer review history of their article (what does this mean?). If published, this will include your full peer review and any attached files.

Reviewer #1: No

Reviewer #2: No

---

## [Author Response · Author response to Decision Letter 0]

19 Aug 2024

Response to reviewers

We extend our sincere gratitude to reviewers for their corrections, contributions, and recommendations, which undoubtedly enhance the quality of our work. All their suggestions have been incorporated into the revised version of the manuscript. Therefore, we will only address the points requiring clarification in the following lines.

Response to reviewer #1:

1. Can you please clarify the univariate and multivariate regression analysis and the variables that are used in the mutlivariate regression analysis? Perhaps a stepwise multivariate regression analysis would be a good approach.

Indeed, we performed a stepwise descending multivariable logistic regression, and the details have been provided in the revised version of the manuscript. Details have been provided in the revised version of the manuscript.

2. Can you please add discussion about limitations such as sampling bias into this study?

A discussion on the potential sampling bias of the study has been added in the strengths and limitations section in the revised manuscript: 

In Benin, according to the latest national surveys, 48% of women of childbearing age did not make at least 4 prenatal visits during pregnancy, whereas WHO recommendations stipulate a minimum of 8 visits. This could introduce a sampling bias in this study; the results may reflect the situation in the general population, however the generalization of the results to the whole country calls for caution.

3. Can you please clarify why you had no follow-up time and why you specifically set the gestational week to be less than 20?

We did not conduct follow-up as the study was a cross-sectional study, not a cohort study. The aim of this study was to determine the frequencies of cardiovascular risk factors before pregnancy that predispose to preeclampsia during pregnancy. According to the recommendations of scientific societies and as specified in the introduction, hypertension or diabetes detected in the first trimester (before 20 weeks of gestation) are due to preexisting chronic disorders. Similarly, in pregnant women, mid-upper arm circumference is a measure that does not vary significantly during the first weeks of pregnancy and can be used to assess overweight and obesity prior to pregnancy.

4. Please include ethic approval numbers and dates

The study was approved by the ethics and health sciences research committee of Parakou University (Benin) on 08th January 2020 (REF:0594/CLERB-UP/P/SP/P/SA). 

We have included this in the revised version of the manuscript.

5. "Blood pressure was measured by an electronic device"- for all women? None were measured manually, perhaps for instance in the setting of atrial fibrillation? Was the electrinic devices calibrated and checked for accuracy?

Blood pressure was measured using an electronic device for all women. We did not perform manual measurements. All measurements were conducted following the recommendations of the STEPwise approach to NCD risk factor surveillance (STEPS). The electronic devices were calibrated, and their accuracy was verified before data collection.

Precisions were included this in the revised version of the manuscript.

6. Conclusion: The frequency of CVRF in women before 20th week of gestation was high. Can you please rewrite and avoid "high" since it is subjective.

“High” was remplaced by “significant” in conclusion.

Response to reviewer #2:

Major comments

1. Page 6; line 119: What is the basis for the definitions of overweight and obesity in Brachial circumference?

The definitions of overweight and obesity in Brachial circumference are based on the study published by Ahminah et al. in 2017 in the South African Medical Journal titled: "Mid-upper arm circumference: A surrogate for body mass index in pregnant women." This definition was used by authors in previous study and in the same context like us.

2. The author should mention the difference between urban and rural backgrounds in the results of Table 1.

The differences between rural and urban settings have been mentioned in materials and methods sections of the study on page 5, lines 97 and 98.

The title of table 1 has been corrected: Table 1: Global sociodemographic characteristics of the women according to area

3. Page 9; line 157: Which is more accurate to say that the patient had normal blood pressure but was receiving treatment or that the patient had normal blood pressure because of the treatment?

In stating 'the patient had normal blood pressure but was receiving treatment,' we aimed to provide information regarding the group of hypertensive patients who were on treatment before the study and had normal blood pressure at the time of the study. The sentence was modified.

4. In Table 2, the definition of the classification of salt intake should be clarified as to how many grams of salt or sodium it is equivalent to.

We are unable to provide information on the grams of salt in the context of our study (African countries with a significant frequency of illiterate women). Available resources did not allow for measurements to be taken. Therefore, data were collected based on the self-reports of patients and according to their perception, as detailed in the table. Advanced nutritional studies in this population indeed seem necessary.

5. In Table 4, it is necessary to indicate which confounding factors were adjusted for.

Confounding factors were adjusted for age and place of residence in the model. This was specified in the revised version of the article (page 6, line 168) and in Table 4.

6. In Table 4, it is necessary to show the relationship between overweight, obesity, smoking, and alcohol consumption as factors associated with hypertension in pregnant women.

We included theses variables in our analysis except for obesity (Supporting information), which was treated separately to avoid confounding biases. However, they did not emerge as associated with hypertension in the univariate analysis and all had p-values greater than 0.20. Therefore, we did not include them in the multivariate analysis or present them in the study, as our primary objective was to provide frequencies of cardiovascular risk factors found by other studies to be associated with preeclampsia (hypertension, diabetes, and obesity). Below are the results of analyses.

- Hypertension and tobacco consumption before pregnancy: P=0.2998

- Hypertension and tobacco consumption during pregnancy: P=1.0000

- Hypertension and alcohol consumption before pregnancy: P=0.2680

- Hypertension and alcohol consumption during pregnancy: P=0.6535

7. Author should mention limitations and strengths in the discussion.

Limitations and strengths were mention in revised versions (page 15, line 237).

Minor comments:

1. Page 9; line 157: Twenty-eight women (2.3%) →28 (2.3%)

The comment has been taken into account.

2. Page 9; lines 161-163: Add (%) after the number of subjects.

The comment has been taken into account.

3. In Table 4, the reference should be shown for each variable

The reference categories have been specified in the tables by indicating a value of 1 for the odds ratios (OR).

---

## [Decision Letter · Decision Letter 1]

2 Sep 2024

High prevalence of cardiovascular risk factors in pregnant women in Benin

PONE-D-24-09850R1

Dear Dr. Lacroix,

We’re pleased to inform you that your manuscript has been judged scientifically suitable for publication and will be formally accepted for publication once it meets all outstanding technical requirements.

Kind regards,

Yoshihiro Fukumoto

Academic Editor

PLOS ONE

Additional Editor Comments (optional):

Reviewers' comments:

Reviewer's Responses to Questions

**Comments to the Author**

1. If the authors have adequately addressed your comments raised in a previous round of review and you feel that this manuscript is now acceptable for publication, you may indicate that here to bypass the “Comments to the Author” section, enter your conflict of interest statement in the “Confidential to Editor” section, and submit your "Accept" recommendation.

Reviewer #1: All comments have been addressed

Reviewer #2: All comments have been addressed

2. Is the manuscript technically sound, and do the data support the conclusions?

Reviewer #1: Yes

Reviewer #2: Yes

3. Has the statistical analysis been performed appropriately and rigorously? 

Reviewer #1: Yes

Reviewer #2: Yes

4. Have the authors made all data underlying the findings in their manuscript fully available?

Reviewer #1: Yes

Reviewer #2: Yes

5. Is the manuscript presented in an intelligible fashion and written in standard English?

Reviewer #1: Yes

Reviewer #2: Yes

6. Review Comments to the Author

Reviewer #1: (No Response)

Reviewer #2: All of my comments in the review have been addressed.

It has been demonstrated that early cardiovascular risk screening for pregnant women is crucial to ensure a healthy pregnancy and childbirth for both mother and child in African communities.

7. PLOS authors have the option to publish the peer review history of their article (what does this mean?). If published, this will include your full peer review and any attached files.

Reviewer #1: No

Reviewer #2: No

---

## [Editor Report · Acceptance letter]

21 Sep 2024

PONE-D-24-09850R1 

PLOS ONE

Dear Dr. Lacroix, 

I'm pleased to inform you that your manuscript has been deemed suitable for publication in PLOS ONE. Congratulations! Your manuscript is now being handed over to our production team.

Kind regards, 

on behalf of

Dr. Yoshihiro Fukumoto 

Academic Editor

PLOS ONE